# Ecological Restoration Plasters and Mineral Pigments Designed with Raw Material from the Island of Gavdos [†]

**Afroditi Fotiou** [1] , **Claire Oiry** [1], **Kali Kapetanaki** [1], **Vassilis Perdikatsis** [2],
**Nikolaos Kallithrakas-Kontos** [2] **and Pagona-Noni Maravelaki** [1],*

[1] MaCHMoB (Laboratory of Materials for Cultural Heritage and Modern Building),
   School of Architectural Engineering, Technical University of Crete, 73100 Chania, Greece;
   afroditi.fotiou@arch.tuc.gr (A.F.); claire.oiry@gmail.com (C.O.); kelly_kapt@hotmail.com (K.K.)

[2] School of Mineral Resources Engineering, Technical University of Crete, 73100 Chania, Greece;
   vperdik@mred.tuc.gr (V.P.); nkallithrakas@isc.tuc.gr (N.K.-K.)

* Correspondence: pmaravelaki@isc.tuc.gr; Tel.: +30-28210-37661

† This paper is and extended version of our paper published in Fotiou, A.; Oiry, C.; Kapetanaki, K.;
   Perdikatsis, V.; Kallithrakas-Kontos, N.; Maravelaki, N.P. Ecological restoration mortars and plasters
   designed with raw material form the island of Gavdos. In Proceedings of 2020 8th Euro-American Congress
   REHABEND, Granada, Spain, 28 September–1 October 2020.

**Abstract:** Gavdos is an island of ca. 34 km$^2$ located to the south of Crete, Greece, with a local landscape rich in clay material of remarkable diversity in colour and quality. The limited natural and human-made resources are persistently recycled, forming the built structures of the island and determining the island's sustainable local tradition. In the framework of this research, areas with clay soil were identified through a geological survey and testimonies of local inhabitants. The studied clay samples were characterized with mineralogical and physicochemical analyses. Two out of ten samples with a clay content higher than 50%, after low-temperature thermal treatment (600 °C and 700 °C), functioned as pozzolanic additives enhancing the performance in resistance to salt decay and plasticity of lime mortars. Seven raw clay samples were used as pigments in lime-based colours and their performance and durability, as assessed with the appropriate laboratory analyses, revealed the existence of stable mineral pigments under UV and visible light irradiation. There is great potential in the exploitation of local raw material from the island of Gavdos for the restoration of the traditional building stock on the island in terms of resource efficiency, environmental impact and preservation of the local identity.

**Keywords:** restoration mortars; calcined clays; low thermal treatment; pozzolanic activity; stable mineral pigment

## 1. Introduction

Gavdos island, located 38.89 km from the south coast of Crete, in the Libyan Sea, is a place of outstanding natural beauty. During its long history, it has experienced periods of flourishing development, such as throughout the Neolithic and Bronze Age, during Hellenistic, Roman and Early Byzantine times, as well as serious decline during the 19th and 20th centuries. A typical characteristic of this remote location throughout the years has been the need for constant recycling of the limited natural and human-made resources [1]. The traditional architecture of the 20th century found on the island is an example of this practice with austere building shapes and the use of local materials. Stone, wood, juniper trunks and earth are the main construction materials used on the

island. Clay-based mortars, clay interior plasters and roof layers were extensively applied in traditional constructions, whereas lime mortars for exterior plasters were introduced after 1940. Today, the use of conventional building materials such as reinforced concrete, cement plastering and acrylic paint threaten irreversibly the traditional building stock to extinction. For the sustainable development of the island and the preservation of its natural beauty, the exploitation of its local resources and their use for the restoration of the traditional buildings is of strategic importance.

The existence of pottery kilns and the surface pottery of excellent quality, dating from the Neolithic age, constitute a significant evidence of the quality of local soil. In particular, kaolinite was identified in some clay soil of good quality; kaolinite transformation to metakaolin by dehydroxylation at 600–800 °C thermal treatment provides mortars with pozzolanic properties. According to Budak et al., 2008, the pozzolanic activity of calcined clays is greatly dependent on the firing temperature [2]. According to Tironi et al., 2011, the kaolinite content of clay is not the only parameter influencing the pozzolanic activity; the structural order/disorder of kaolinite before calcination and the specific surface area after calcination also play a distinctive role [3]. The advantages of pozzolanic additives for lime mortars have been known since antiquity. Pozzolans can be natural or artificial and have the ability to react with portlandite (Ca (OH)$_2$) in the presence of water, yielding calcium silicate hydrate (CSH), responsible for improved mechanical properties for mortars [2,4,5] and durability in moisture conditions [6]. According to Sabir et al., 2001, an increase in compressive strength of up to 40% was obtained for mortars containing 15% metakaolin [5].

The use of calcined clay as a pozzolanic additive for mortars and concrete has attracted increased attention in recent decades [5]. Metakaolin/lime mortars are highly recommended as restoration mortars on historic masonries because of their chemical and physico-mechanical compatibility with the existing substrate [6]. Calcined clays used as supplementary cementitious materials also have the huge potential to reduce clinker content in cement, with reduced $CO_2$ emissions and similar or even improved mechanical properties [7]. In the present study, a representative number of samples from the island of Gavdos were characterized before and after thermal treatment. Subsequently, they were tested and evaluated (raw or fired) as an additive in lime-based restoration mortars. Two samples from the total of samples analysed presented remarkable pozzolanic properties after thermal treatment in low-temperature firing [8].

The local clay soils were also used unfired as pigments in lime-based colours. The technique of mural, painting with lime-based colours on lime plaster, has a long history. The first murals were found in the Temple of Tepe Gawra (3500 BC) in Iraq and in the Royal Palace of Mari (2000 BC) in Syria. Fragmentary remains of fresco wall paintings of the Minoan settlement at Knossos on Crete from almost every period from Early Minoan I to Late Minoan IIIB (3000–1200 BC) have been revealed from the stratigraphical excavations of the British School at Athens conducted after the Second World War [9]. The lime-based colours are applied "a secco" (on dry surface), "a mezzo fresco" (on semi fresh surface) or "a fresco" (on fresh surface). The lime-colours used in the present study are limewater applied "a mezzo fresco" (24 h after plaster application) and limemilk applied "a secco" [10].

## 2. The Island of Gavdos

Gavdos constitutes a diverse landscape, offering anchorages along the north, east and south coasts and an irregular terrain, rising up to 368 m [11]. The Mediterranean coastal climate of the island is characterized by warm wet winters and hot dry summers [12].

### 2.1. Historical Context

The island of Gavdos presents a consistent human activity from the early Paleolithic period, as evidenced by a large number of relevant stone artefacts found on the island. Surface pottery of very good quality scattered through the island is attributed to all the phases of Neolithic period and bares similarities to those identified in central Crete for the same period. From the Hellenistic period onwards, the main settlement on the island was located on the hill of Ai Yiannis, on the north coast

and the bay of Lavrakas, northwest of the hill, was used as a harbor. During the Roman period, the rural economy on the island flourished, influencing various sectors, such as craft, industries and infrastructure works, an outdoor stone-built winemaking installation on the edge of Ayios Pavlos stream and an aqueduct supplying water to the Ai Yannis settlement are representative findings of that period. According to the archaeological survey, the settlement of Ai Yannis continued to flourish at least up to the early Medieval period. During the Ottoman conquest on the island the Lighthouse was built; it consisted of a circular tower of 14 m in height attached to the keeper's house. In the beginning of the 20th century, the island was reported to be inhabited by less than 50 families and the countryside appeared to be deserted. After 1930, Gavdos received its first political exiles who constructed their shelter themselves. The building was built in local traditional style with a central beam from a juniper tree supporting the clay roof [13].

## 2.2. Geological Information

The island of Gavdos has a wide variety of geological formations. The oldest one belongs to the Unit of Kalypsos, which is restricted to the northeast edge of the island. The Unit of Pindos represents the alpine background of Gavdos and is developed along the southwest shore and in a narrow zone along the east coast of the island, containing flysch and limestone. The neogene formations cover 2/3 of the island; they consist of clays, grey-blue to grey-green and yellow-white marl, marly limestone, calcitic sandstone, fine-grained sand and sandy clay [14]. The youngest formations are found scattered across the island; they mainly consist of pleistocenic marly limestone, aeolianites, sand and dunes (as shown in Figure 1). The raw material of interest for the present research is clay. Clay resources are available in the Unit of Pindos, in neogene and pleistocenic sediments and in red soils produced from the erosion of carbonate rocks and the flysch of Pindos [15].

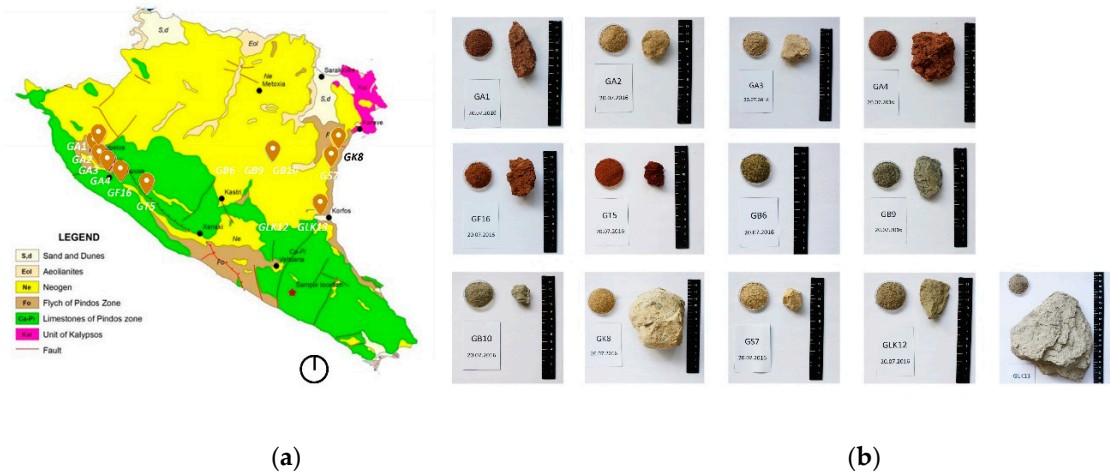

(**a**)　　　　　　　　　　　　　　　　　　　　　　　(**b**)

**Figure 1.** (**a**) Map of geologic formations of Gavdos [15]; sampling positions and (**b**) colour palette of the samples.

## 2.3. Traditional Architecture

Farmsteads, locally known as "Metochia", are the dominant type of architecture on the island. They are built in the traditional Cretan folk style with a simple form, restricted to what is needed. They were used not only for agricultural activities, but also as homes. The construction materials used were local rough stone, juniper trunks and branches for the roof, covered with clay. Traditional buildings had interior plaster from clay and finely chopped straw, and exterior lime-based plastering was introduced after 1940 [13]. The traditional building stock of the island is in urgent need of protection from the conventional restoration methods applied with reinforced concrete, cement plastering and acrylic paint [16]. Dense, low water-vapour-permeable cement mortars are incompatible with the traditional materials and cause irreversible damage on the authentic substrate [17]. Furthermore,

replacement of the flexible plaster roof with horizontal reinforced concrete slab jeopardize structural safety by reducing maximum displacements and plastic strains in certain areas [18].

## 3. Materials and Methods

In the present study, the use of local clay soil was tested according to the methodology presented in detail in the following subsections.

### 3.1. Identification of Clay Soil—Characterisation of Samples

The first step was to determine the sampling positions. The criteria set for this procedure was to find sources of good quality clay soil, not located in the special protected area LIFE Natura 2000 and available for extraction in larger quantities (1–2 m$^3$). The sampling positions were chosen according to the geological survey and testimonies of locals. A total of 13 samples were taken from the areas of Amplelos, Trohaliaris, Vardaris, Klimousena, Ayioi Pateres, Lakos Korfou and Lakos Farou. The sampling areas in the island are depicted in Figure 1 in Section 2.2.

The texture, as well as the physico-chemical and mineralogical properties of samples were revealed with the aid of specific techniques. The wash hand field test was primarily used to rule out the samples with no sufficient clay content that were easily removed with soft washing, without leaving any traces of clay on the hands. High-plasticity clays tend to feel greasy and are difficult to wash out in comparison with low-plasticity clays and silts [19]. The characterisation of samples proceeded with the biscuit test, with soil and water, which determines the shrinkage, abrasion and cohesion of samples [20]. An Energy Dispersive X-Rays Fluorescence (EDXRF) analysis, performed with Spectro Xepos from Ametek, gave insight into the variations of the composition of major elements in each sample [21]. An X-Ray diffraction (XRD) analysis (Bruker D8 advance diffractometer) was performed in most of the specimens in order to identify the mineral crystalline phases and to complete the mineralogical characterisation. The quantitative XRD analysis was carried out using the Rietveld Method with the TOPAS Program from Fa. BRUKER. A differential thermal analysis-thermogravimetric analysis (DTA-TG) was performed with Setaram LabSys Evo 1600 °C thermal analyser up to 1000 °C in order to determine the temperature range of the endothermal dehydroxylation of clay [22].

### 3.2. Thermal Treatment—Assessment of Hydraulic Properties—Pozzolanicity

According to Budak et al., 2008, the pozzolanic activity of fired clays is greatly dependent on the firing temperature and is more intense in clays with low calcite and high kaolinite content [2]. Kaolinite is a two-layer electrically neutral clay with a low specific surface area, thus allowing the low penetration of water [23]. In the present study, the selected samples were heated in a box kiln at temperatures of 600 and 700 °C for 3 h. The pozzolanic activity of the clay samples was assessed with the Luxan Method. This analysis involves the measurement of the electrical conductivity of samples before and after reacting with saturated calcium hydroxide solution. Samples exhibiting an electrical conductivity difference between 0.4 and 1.2 mS cm$^{-1}$ demonstrate moderate pozzolanicity, whereas samples with an electrical conductivity difference higher than 1.2 mS cm$^{-1}$ demonstrate good pozzolanicity [24].

### 3.3. Synthesis of Mortars with Calcined Clay Soil

The binders selected for the synthesis of mortars were lime putty with pozzolanic additives. The samples that demonstrated the best results in the previously documented procedure were used as additives after thermal treatment in low temperature. Mortars 01-GF16 and 02-GF16 contain calcined clay (at 700 °C) from Gavdos "G" from the area near the Lighthouse "F" (Faros in Greek), whereas 01-GT5 contain calcined clay (at 600 °C) from Gavdos "G" from the area of Troxaliaris "T". Numbers 5 and 16 are the serial number of each sample. In Figure 1, the sampling positions and the name of each sample are outlined. Additionally, pozzolanic products from the national market (pozzolan and zeolite) were added in mix designs to serve as a reference. Mortar 00-A contains

pozzolan and mortar 00-B zeolite. In one synthesis, Natural Hydraulic Lime (NHL 3.5) had partially substituted lime putty for the production of a mortar with enhanced mechanical properties and water resistance (as shown in Table 1). The selected aggregate was standard sand with grains ranging from 0–4 mm from a Cretan quarry. Table 1 lists the mix design of mortars. Thermally treated soil with granulometry up to 2 mm was mainly used for the preparation of specimens because it is easier to be produced in situ. The products from the market have significantly finer granulometry favouring the reaction with lime and the hardness acquired; therefore, for comparison purposes, a lower proportion of additives were used compared to the syntheses with thermally treated soil. An additional criterion for the selection of the materials was their local origin and affordable price. The lime putty was acquired from the lime production company ASBEK in Fones, Chania, Crete, pozzolan from PROLAT company in Xyropigado—Mandra, Attica and zeolite from Olympos Company in Assiros, Lagadas.

**Table 1.** Proposed mix designs.

| Name | Synthesis | | | |
|---|---|---|---|---|
| | Lime Putty (% *v/v*) | NHL 3.5 (% *v/v*) | Sand (0/4) (% *v/v*) | Additive (% *v/v*) |
| 00-A | 16.4 | - | 73.8 | 9.8 (pozzolan PROLAT, 0–75 μm) |
| 00-B | 16.4 | - | 73.8 | 9.8 (zeolite Olympos, 0–45 μm) |
| 01-GF16 | 25 | - | 60 | 15 (GF16 700 °C, 0–2 mm and 0–63 μm) |
| 01-GT5 | 25 | - | 60 | 15 (GT5 600 °C, 0–2 mm) |
| 02-GF16 | 18.5 | 7.4 | 66.7 | 7.4 (GF16 700 °C, 0–2 mm) |

### 3.4. Quality Control of Mortars with Calcined Clay Soil

Restoration mortars for the conservation of traditional and historic buildings should have a mechanical and physical performance compatible with the existing substrate and allow retreatability in order not to compromise future interventions [25]. The analyses carried out assessed the performance of the produced mortars in properties that address their role in the construction. From the implemented analyses the mechanical properties, water absorption, shrinkage and resistance in salt crystallization were tested and evaluated. In Table 2, the implemented laboratory analyses are presented. For every test, three mortars from each mix design are used, with the exception of shrinkage test, with only one specimen from each mix design and uniaxial compressive strength, with six specimens, three for 28 days' curing time and three for 3 months' curing time. Figure 2 illustrates the types of specimens used for this study. The values reported in the following sections represent the average values of three samples for each test.

**Table 2.** Laboratory analyses for proposed mortars.

| Name of the Test | Type of Specimen | No of Specimens | Curing Time | Standard |
|---|---|---|---|---|
| Shrinkage test | Biscuits of 4.5 cm diameter and 1 cm thickness | 1 × 5:5 | 1 week | - |
| Uniaxial compressive test | 50 × 50 × 50 (mm) | 3 × 5:15 3 × 5:15 | 28 days and 3 months | - |
| Capillary water absorption | Cylindrical specimens 100 mL | 3 × 5:15 | 3 months | UNI-EN 15801:2010 |
| Crystallization of salts | Cylindrical specimens 100 mL | 3 × 5:15 | 3 months | EN 12370 |

For the shrinkage test, three cylindrical specimens (biscuits) were prepared for every mix design presented in Table 1, with the use of a ring-shaped mould. The specimens dry in shadow without the mould. After a week the specimens are placed again in the mould and the shrinkage percentage is calculated [20].

The determination of compressive strength of hardened mortars is carried out in cubic specimens of 50 × 50 × 50 mm by partially modifying the EN 1015-11:2019 [26]. The specimens remained in the mould for 3 days and then they were stored in curing chamber with stable temperature and humidity. Because the specimens contain aerial lime that hardens with air, they remained 2 weeks in and 2 weeks outside the chamber (for 28 days curation) and 1 month in and 2 months outside the

chamber (for 3 months curation). This methodology was followed because the authors noticed that it resulted in harder specimens. For each test, three specimens are required. During the test, the specimen is placed among two loading plates and compressed until failure with simultaneous recording of the load enforced and the provoked deformation in mm.

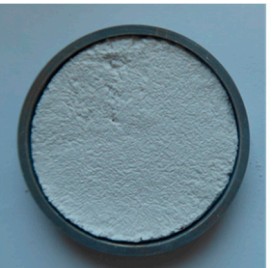 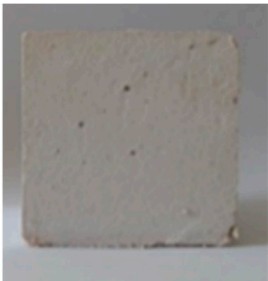 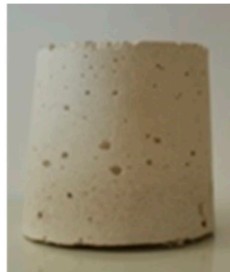

**Figure 2.** Type of specimens for the laboratory analyses—biscuit, cube, cylindrical specimen.

The capillarity test determines the amount and rate of water absorption through the test surface in contact with water; the experimental procedure was implemented according to the standard UNI-EN 15801:2010 [27]. The following procedure was adopted to determine the capillarity water absorption coefficient. The specimens were dried to constant mass in a ventilated oven at a temperature of 60 °C for 48 h, then they were placed in a desiccator until they reached room temperature. Subsequently, they were weighed and placed on a saturated bedding layer with 5 mm thickness. The bedding layer was kept saturated by adding distilled water when necessary and weighing of the specimens was implemented in short intervals for 48 h.

The test for crystallization of salts aims to define the resistance of the specimens to the crystallization of sodium sulfate salt. For this test, three cylindric specimens of 100 mL from each mix design were dried in a ventilated oven and subjected to 15 cycles of salt crystallization. Each cycle includes weighing of the specimen, placement on bedding layer saturated with a solution of 100 g $H_2O$ and 14 g $NaSO_4$ for two hours. In addition, the bedding layer is kept saturated by adding the solution when necessary, placement of the specimen in ventilated oven for 10 h at a temperature of 100 °C, weighing and photographing of the specimen are performed.

### 3.5. Colour Palette of Clay Soils

The ancient Greek painter Polygnotus established the use of four basic colours and played a distinctive role in the development of the classical art of painting. The use of the tetrachromy of Polygnotus provided the inspiration for the present study, as it was perceived as a choice and not a necessity and was completely harmonized with the Greek landscape combined with the plenty and vivid light [28]. The clay soils of Gavdos island, in addition to their high content of good quality clay, also possess a wide palette of unique colours. These colours form part of the natural and built environment of the island and therefore, cannot be omitted from the proposal for a sustainable conservation of the traditional building stock of the island. A study on the existing colour palette of the island was performed and provided the basis for the development of the colour palette with the use of local clay soils.

The local clay soil was then tested as pigment in lime colours and more precisely in the production of limewater and limemilk. These two techniques of lime paint production offer an adequately intense aesthetic result in order to identify in it the colour of the earth and yet are discrete and in harmony with the colouration of the traditional structures on the island. The difference between these two techniques lies in their lime and earth content. The coloured limewater has a lime and water ratio of 1:6 and 65% (*v/v* %) earth and the coloured limemilk has a lime and water ratio of 1:3 and 25% (*v/v* %) earth.

### 3.6. Synthesis of Lime-Based Colours with Clay Soils

The methodology followed for the production of lime colours is summarized below:

- Determination of the volume of lime and water to be used according to the recipe (limewater or limemilk)
- Heating of a small quantity from this water and impregnating the earth for 1–2 days
- Mixing all the ingredients together and letting the limecolour mature for 2 weeks
- Applying the limewater (1:6) in "semi fresco" plaster (24 h after the implementation of the plaster) and the limemilk (1:3) in "secco" plaster after having sprinkle the surface
- The application of limecolours is implemented in three layers with 4 h interval between each layer.

Every colour and lime colour technique was afterwards tested with four different stabilizers. The stabilizers used were casein [29], potassium alum [30], nopal juice [31] and linseed oil [32]. Table 3 lists the different mix designs of limecolours with stabilizers. With 7 different pigments (clay soils) and 10 different mix designs, a total of 70 different limecolours were produced and applied on biscuits and on an exterior wall surface for the implementation of the quality control tests.

**Table 3.** Synthesis of limecolours.

| Name | Synthesis | | Clay Soil (% *w/w* of Lime) | Stabilizer |
|------|-----------|-----------|------------------------------|------------|
| | Water (% *v/v*) | Lime (% *v/v*) | | |
| A-0 | 75 | 25 | 25 | - |
| A-1 | 75 | 25 | 25 | 1% *v/v* casein |
| A-2 | 75 | 25 | 25 | 50 gr potassium alum for 6 L water |
| A-3 | 75 * | 25 | 25 | * nopal juice instead of water |
| A-4 | 75 | 25 | 25 | 3% *v/v* linseed oil |
| B-0 | 86 | 14 | 65 | - |
| B-1 | 86 | 14 | 65 | 1% *v/v* casein |
| B-2 | 86 | 14 | 65 | 50 gr potassium alum for 6 L water |
| B-3 | 86 * | 14 | 65 | * nopal juice instead of water |
| B-4 | 86 | 14 | 65 | 3% *v/v* linseed oil |

### 3.7. Quality Control of Lime-Based Colours with Clay Soils

The produced limecolours with or without natural stabilizers were subjected to a number of laboratory tests to assess their strength and durability. In Table 4, the implemented laboratory analyses are presented. Each of the 70 different limecolours was applied in three biscuits (as shown in Figure 3a) for the implementation of UV and adhesion tests. The abrasion and sponge test were implemented only for the syntheses with stabilizers that were applied on an exterior wall surface (as shown in Figure 3b). The two type of surfaces used for this study are presented in Figure 3.

**Table 4.** Laboratory analyses for developed limecolours.

| Name of the Test | Type of Specimen | No of Specimens | Curing Time | Standard |
|------------------|------------------|-----------------|-------------|----------|
| UV-colorimeter | Biscuits (4.5 cm) | 3 × 70:210 | - | - |
| Adhesion (scotch tape) test | Biscuits (4.5 cm) | 3 × 70:210 | 30 days | - |
| Abrasion test | Exterior wall surface | 8 × 7:56 (only syntheses with stabilizers) | 7 days | - |
| Sponge test | Exterior wall surface | | 30 days | UNI 11432:2011 |
| Water vapour transmission (cup method) | Biscuits (4.5 cm) | 5 | 30 days | EN ISO 12572:2016 |

The UV test determines the stability of colours after the exposure in UV radiation. Every colour was applied in two specimens (as shown in Figure 4); the first specimen was placed in a UV chamber, with four blacklight lamps T8-15W, and the second in a dark room. Every 7 days, the colour difference

(ΔE) from the previous measurement was measured with a colorimeter in three specific spots of each specimen. The test lasted 28 days.

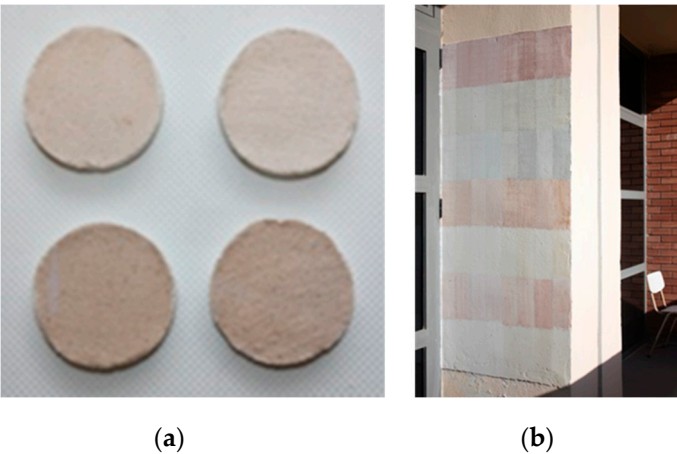

|         |         |
|---------|---------|
| (**a**) | (**b**) |

**Figure 3.** Type of specimens for the laboratory analyses (**a**) biscuits, (**b**) exterior wall surface.

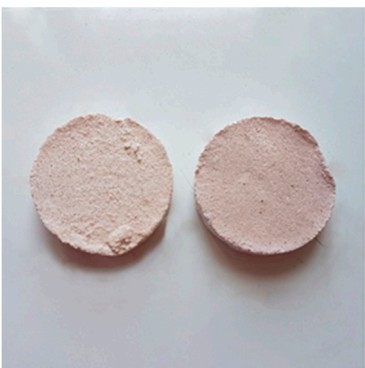

**Figure 4.** Specimens used for the UV test.

According to the American Society of Testing and Materials (ASTM), there are five lightfastness categories that determine the colour stability based on the colour difference before and after placement in the UV chamber [33–35]:

- Lightfastness I: Excellent performance (ΔE < 4)—suitable for outdoor application
- Lightfastness II: Very good performance (8 < ΔE < 4)
- Lightfastness III: Fair performance (16 < ΔE < 8)
- Lightfastness IV: Poor performance (24 < ΔE < 16)
- Lightfastness V: Very poor performance (24 < ΔE)

The adhesion test assesses the power with which the colour is attached to the substrate. For the implementation, a scotch tape is weighed with a precision of 2 decimal places, before and after being stuck on the coloured surface (as shown in Figure 5). The adhesion test was implemented in two specimens of each mix design. The adhesion test assesses the contribution of the natural stabilizers in the stability of colour imposed in mechanical stress [36].

An abrasion test is complementary to adhesion test and determines the durability of colours in mechanical friction. In this test, a cotton swab is rubbed against the colour for a distance of 10 cm three times (up-bottom, bottom-up, up-bottom) (as shown in Figure 6). The swab is weighed before and after the application.

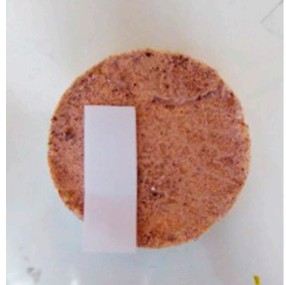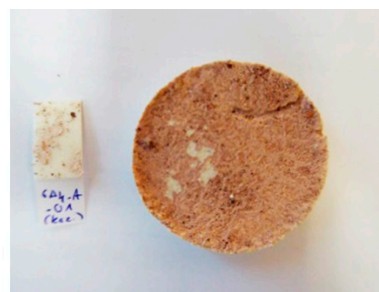

**Figure 5.** Adhesion test (scotch tape).

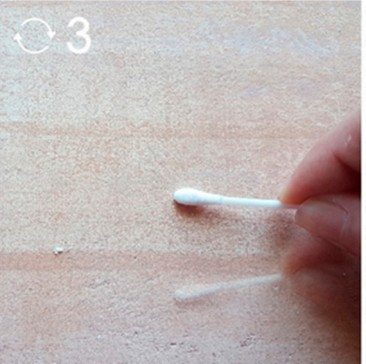

**Figure 6.** Abrasion test.

The sponge test (absorption test) is a non-destructive test that determines the absorption capacity of the coating material. During the test, a wet sponge with a 6-cm diameter is weighed, pushed against the surface with light pressure for 30 s and then weighed again (as shown in Figure 7) [37].

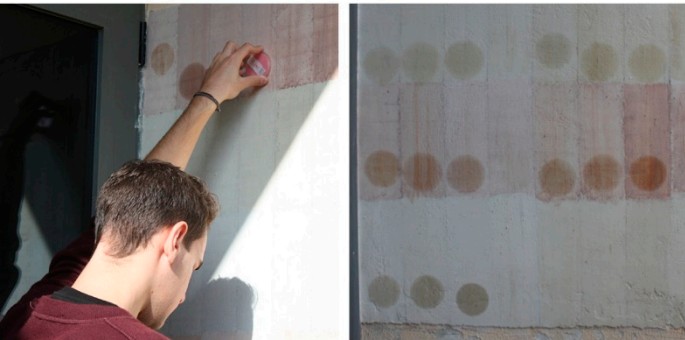

**Figure 7.** Sponge test (absorption test).

The water vapour transmission test determines the water vapour permeance of a specimen and the water vapour permeability under isothermal conditions. The test was carried out according to the guidelines of the CEN, EN ISO 12572:2016 standard procedure [38]. More specifically, in the water vapour transmission test, the specimen is placed and sealed on a test cup above a pot of distilled water. The assembly is placed in a temperature- and humidity-controlled test chamber. Because of the different partial vapour pressures between the test cup and the chamber, a vapour flow occurs through permeable specimens. Periodic weighing of the assembly is made to determine the rate of water vapour transmission in a steady condition. The water vapour transmission test lasted two weeks [38,39].

## 4. Results

### 4.1. Sampling Areas of Clay Soils

Based on the geological survey and testimonies of inhabitants, a total of 13 samples were collected from areas outside of the special protected area LIFE Natura 2000, where the acquisition of a bigger quantity, 1–2 m$^3$ of soil, would not affect the natural landscape. More specifically, samples were collected from the red pelites and radiolarites of Pindos zone, south of Korfos Bay and north from the lighthouse in Ampelos, from brown-green or blue marles found in the sediments of flysch of Pindos zone, from neogene formations of grey-blue to grey-green marles from Ayioi Pateres, Vardaris, Klimousena, Korfos and from red soil in the areas of Ampelos and lighthouse. The sampling positions, along with the aesthetic properties of the samples are illustrated in Figure 1. Every sample received a distinctive code formed by the letter "G" for Gavdos, followed by the first letter of the location from which the sample was acquired, for example "A" for Ampelos and the serial number of the sample.

### 4.2. Characterisation of Samples

During the first field test, specifically the wash hand test, a first estimation of the content and quality of clay in the samples was accomplished. A total of 10 samples were chosen, namely GA1, GA3, GA4, GT5, GF16, GB6, GB9, GLK12, GK8 and GS7, on the basis of their high clay content, good quality of clay and the uniqueness of their colour (for example GT5, GB9). A simple soil identification test, such as the biscuit test with soil and water, was subsequently performed to determine shrinkage, cohesion and abrasion of samples. The samples GB6, GB9, GF16, GLK12 and GT5 exhibited a high percentage of shrinkage, an indication of very reactive clay. According to an XRD analysis, the samples GF16 and GT5 contained the highest percentage of clay, namely 71.23% and 83.70%, respectively. On the other hand, the samples GB6, GB9 and GLK12, contain Gypsum, which is also considered to be responsible for swelling and shrinkage phenomena. The least coherent and non-resilient to abrasion samples were GA1 and GA3, which both have high Calcite content.

The elemental content of each sample outlined by the XRF analysis is shown in Table 5, whereas the mineral compounds determined by the XRD patterns are shown in Table 6. According to XRF analysis the main elements present in the samples are Silicon, Calcium, Aluminum and Iron. The samples with higher Silicon than Calcium content are GA1, GA4, GT5, GF16, GB6, GB9 and GK8. XRD analysis was performed for the 10 samples selected as most appropriate from the wash hand tests; the XRD patterns indicated the presence of calcite, illite, quartz, kaolinite, chlorite and albite in all of the samples. In particular, the samples GB6, GB9 and GLK12 contain Gypsum as well. Combining the results of XRF and XRD, it can be observed that the sample GT5 has the highest concentration in Silicon and Kaolinite as opposed to GA3 showing the highest Calcium and Calcite content.

**Table 5.** XRF analysis of samples (% *w/w*).

| XRF | GS7 | GA1 | GA3 | GA4 | GB6 | GB9 | GF16 | GK8 | GLK12 | GT5 |
|-----|-----|-----|-----|-----|-----|-----|------|-----|-------|-----|
| Mg | 1.15 | 1.08 | 1.31 | 1.40 | 2.89 | 3.40 | 1.89 | 2.14 | 3.19 | 1.13 |
| Al | 3.74 | 4.01 | 3.16 | 6.25 | 6.30 | 6.87 | 8.83 | 5.68 | 5.13 | 10.35 |
| Si | 10.47 | 18.78 | 8.45 | 19.16 | 17.56 | 18.61 | 20.78 | 16.91 | 14.50 | 21.13 |
| Ca | 24.96 | 17.55 | 28.57 | 10.56 | 13.07 | 10.36 | 5.13 | 13.63 | 16.74 | 1.40 |
| Fe | 3.10 | 3.54 | 2.09 | 5.41 | 4.52 | 4.74 | 5.74 | 4.53 | 3.61 | 6.27 |

Soil samples underwent heat treatment at 600 and 700 °C for 3 h. After calcination, the raw samples of grey, green and yellow colour obtained an orange-reddish hue (as shown in Figure 8). DTA-TG of the raw samples GA4, GT5 and GF16 corroborated the results of XRF and XRD analyses. The first endothermic peak for all samples appears in the temperature range of 100 °C and corresponds to the hygroscopic water captured into the samples. The second endothermic peak, in the temperature range of 500 °C, is related to the clay dehydroxylation, indicating that GT5 has the highest content in

clay. GT5 presents also an intense exothermic reaction at 950 °C–1000 °C, which is associated with the transformation of metakaoline. The endothermic peak at 800 °C is related to the decomposition of $CaCO_3$ and indicates the high content of calcite for GA4 and its absence in sample GT5 (as shown in Figure 9) [40,41].

**Table 6.** XRD analysis of samples: minerals and quantitative analysis (% *w/w*).

| XRD | GS7 | GA1 | GA3 | GA4 | GB6 | GB9 | GF16 | GK8 | GLK12 | GT5 |
|---|---|---|---|---|---|---|---|---|---|---|
| Quartz | 13.4 | 28.3 | 7.3 | 25.3 | 15.9 | 14.5 | 19.4 | 17.4 | 14.4 | 15.3 |
| Calcite | 59.7 | 43.0 | 68.3 | 23.6 | 30.2 | 18.3 | 9.4 | 34.9 | 37.5 | 1.0 |
| Illite | 13.3 | 14.7 | 10.3 | 24.0 | 18.7 | 21.8 | 49.0 | 22.6 | 14.9 | 25.8 |
| Kaolinite | 4.3 | 3.8 | 3.7 | 8.2 | 2.1 | 3.4 | 8.2 | 6.0 | 2.1 | 50.2 |
| Chlorite | 2.6 | 4.3 | 5.4 | 9.6 | 15.2 | 22.4 | 6.8 | 8.0 | 7.3 | 4.9 |
| Albite | 6.7 | 5.8 | 5.1 | 9.4 | 15.9 | 13.3 | 7.3 | 11.0 | 7.4 | 2.8 |
| Gypsum | - | - | - | - | 2.0 | 6.3 | - | - | 16.4 | - |

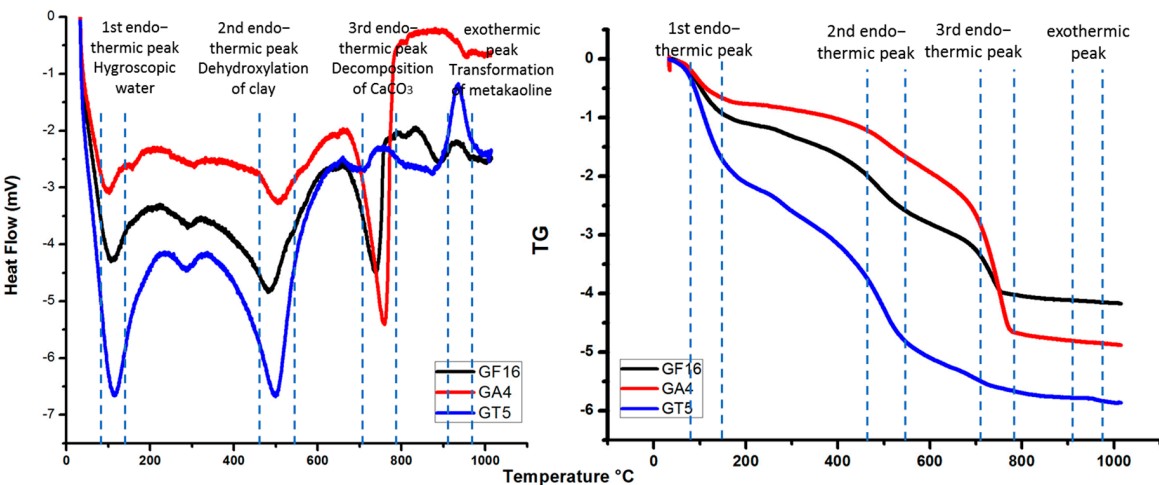

**Figure 8.** Colour change after heat treatment for samples GLK12, GS7, GA3 (upper, left to right), GK8, GB9, GB6 (down, left to right).

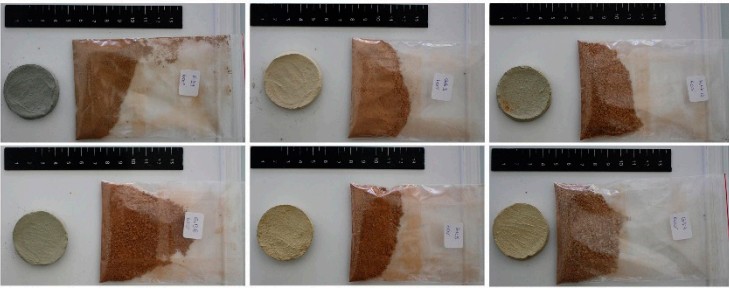

**Figure 9.** DTA-TG analysis of GA4, GT5 and GF16.

### 4.3. Assessment of Hydraulic Properties—Pozzolanicity

Finally, the pozzolanic activity of samples GA4, GT5 and GF16 was evaluated with the Luxan test, measuring the electrical conductivity variation in saturated Ca (OH)$_2$ solution [24]. The results of this test reveal variable pozzolanicity, between 0.4 and 1.2 mS cm$^{-1}$, for the calcined sample GT5 in both 600 and 700 °C and for the calcined sample GF16 only in 600 °C. Namely, GT5, the sample with the highest content of clay (83.7%), and specifically the highest kaolinite (50.2%) along with the lowest content of calcite (1%), demonstrated both in 600 and in 700 °C a mean conductivity difference equal to 0.58 mS cm$^{-1}$. The sample GF16, which has also a high clay (71.2%) and low calcite content

(9.4%), demonstrated a mean conductivity difference equal to 0.9 mS cm$^{-1}$ in 600 °C, without an appreciable conductivity difference for the sample heated at 700 °C. This can be attributed to the lack of pozzolanic properties in overheated fired clays [2], but further testing needs to be performed in different calcination temperatures from 550 to 750 °C. Negative conductivity measurements in the GA4 sample are attributed to its modest calcite content of 23.6%.

Figure 10 presents graphically the procedure followed for the identification of clay soils suitable to be used as hydraulic additives in lime mortar production.

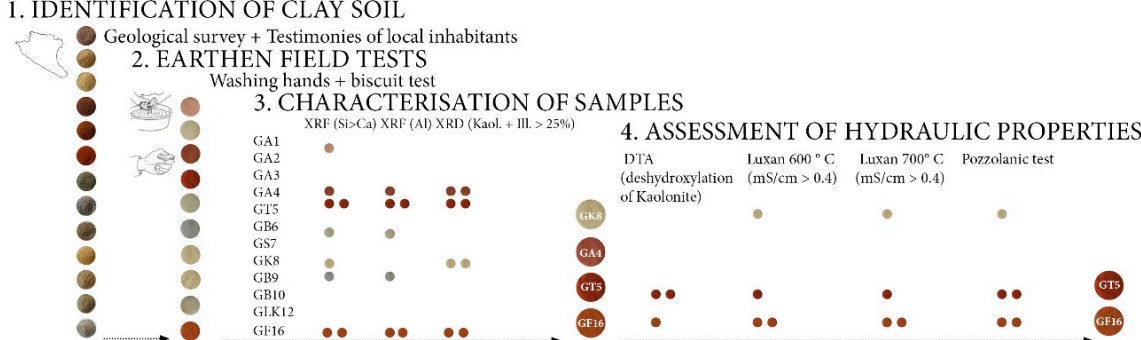

**Figure 10.** Overview of the analysis process.

### 4.4. Quality Control of Mortars with Calcined Clay Soil

For the shrinkage test, the cylindrical mould used to create the biscuits was placed again around the specimen after one week of drying. The higher the shrinkage percentage of a specimen, the higher the probability of cracking is. For each mix design, one biscuit was produced and the shrinkage percentage is presented in Table 7. For the specimens with the mix designs presented in Table 1, there was not significant shrinkage observed.

**Table 7.** Shrinkage percentage.

| Name | % Shrinkage |
|---|---|
| 00-A | 2.17 |
| 00-B | 2.17 |
| 01-GF16 | 2.17 |
| 01-GT5 | 3.26 |
| 02-GF16 | 3.26 |

The results of the uniaxial compressive strength test, Young modulus and maximum deformation are presented in Table 8. The specimens A and B, with products from the market instead of thermally treated soil, were tested as a reference. For every mix design, three specimens were developed and in Table 8, an average value from the three specimens and the standard deviation are presented. For the mix design 01-GF16 28d, two different recipes were attempted: the first one with thermally treated soil with grain size <2 mm and the second one with thermally treated soil with grain size <63 μm. The sample 01-GF16 3 months contained mixed clay grains of 63 μm and 2 mm (65% and 35%, *v/v* of the total clay volume, respectively).

The thermally treated clay from Gavdos of fine granulometry (63 μm), namely 01-GF16, exhibited an almost double compressive strength in 28 days (01-GF16 28 days (63μm)) comparing to the corresponding specimens of thicker granulometry 01-GF16 (2 mm). The Young modulus of the heated clay specimens with fine grains is also three times higher in comparison to the specimens of clay additive with grains up to 2 mm of diameter. Similar findings were also valid for the mortar of clay additive with both fine and thick grains, after a three-month curing (01-GF16 3 months), showing increased values of compressive strength and deformation and stable Young modulus. The other thermally

treated clay with grains up to 2 mm, namely 01-GT5, attains considerable increases in compressive strength, Young modulus and maximum deformation in three months of curing from early stages.

**Table 8.** Compressive strength, Young modulus and Max Deformation for mortars with thermally treated soil from Gavdos (GF16 and GT5) and commercial pozzolanic additives (A and B).

| Name | Compressive Strength (MPa) | Young Modulus (GPa) | Max Deformation (%) |
|---|---|---|---|
| 00-A 28 days | 0.71 (±0.080) | 0.07 (±0.0007) | 2.0 (±0.00) |
| 00-A 3 months | 3.50 (±0.063) | 0.42 (±0.135) | 2.0 (±0.00) |
| 00-B 28 days | 2.29 (±0.081) | 0.40 (±0.075) | 2.0 (±0.00) |
| 00-B 3 months | 3.16 (±0.247) | 0.23 (±0.076) | 2.7 ( ± 0.01) |
| 01-GF16 28 days (2 mm) | 0.68 ( ± 0.005) | 0.05 (±0.005) | 3.0 (±0.00) |
| 01-GF16 28 days (63 µm) | 1.36 (±0.021) | 0.16 (±0.044) | 2.3 (±0.01) |
| 01-GF16 3 months (63 µm–2 mm) | 1.25 (±0.061) | 0.05 ( ± 0.003) | 4.3 (±0.01) |
| 01-GT5 28 days | 0.69 (±0.020) | 0.06 (±0.010) | 3.0 (±0.00) |
| 01-GT5 3 months | 1.30 (±0.006) | 0.07 (±0.009) | 4.0 (±0.00) |
| 02-GF16 28 days | 0.53 (±0.046) | 0.02 (±0.005) | 4.3 (±0.01) |
| 02-GF16 3 months | 1.32 (±0.093) | 0.06 (±0.003) | 4.0 (±0.00) |

An interesting result derives from the comparison between the sample 02-GF16 containing lime, natural hydraulic lime and half the quantity of clay compared to the other samples with double the quantity of clay additive. More specifically, in 3 months, the mortar with NHL shows similar properties to the GT5 clay additive, whereas in 28 days, a decrease in compressive and Young modulus and better deformation. It is reasonable to assume that a substitution of half of the heated clay quantity with NHL does not impart to mortar improvement of the mechanical properties.

The commercial samples both with fine pozzolan (<75 µm) and zeolite (<45 µm), labelled 00-A and 00-B, respectively, exhibited an increase in the mechanical properties over time, except for a decrease in Young modulus by half in the sample with the finer zeolite. The latter, however, in a 3-month curing period showed better plasticity than the sample with pozzolan, due to the fine grains of pozzolanic additives that could effectively react with lime to promote the hydraulic product formation, imparting plasticity to mortars. It is important to highlight that the designed mortars with clay from Gavdos, comparing to the commercial ones, exhibit double the maximum deformation both in early and longer stages of curing. This is an advantage of employing the selected thermally treated clays in temperature up to 700 °C, as additives in mortars entailed absorbing external loading, due to the afforded considerable values of deformation.

The results of the capillary water absorption test are presented in Figure 11 and Table 9. As can be seen from the capillarity water absorption coefficient, a lower water absorption coefficient appears for the mortar 02-GF16 with NHL 3.5, which also presents the maximum water absorption, the second lower coefficient appears for mortar 01-GT5. The lowest water absorption appears for mortar 00-A. Mortars with products from the market present a higher capillarity water absorption coefficient, 00-A has the same capillarity coefficient with 01-GF16 and 00-B has a higher capillarity coefficient.

**Table 9.** Average capillarity water absorption coefficient

| Name | Capillarity Water Absorption Coefficient (g/(cm$^2$ $\sqrt{s}$)) |
|---|---|
| 00-A | 0.020 |
| 00-B | 0.024 |
| 01-GF16 | 0.020 |
| 01-GT5 | 0.018 |
| 02-GF16 | 0.017 |

The results on the crystallization of salts reveal that samples 00-A, 01-GF16 and 02-GF16 present the highest resistance. In Figures 12 and 13, the curves of % weight loss (ΔB) within the 15 cycles

and the photographical documentation of samples 01-GF16, sample with good behaviour in the crystallization of salts, as well as 01-GT5, samples with moderate performance in the crystallization of salts, are depicted. In Table 10 the average weight loss % for the specimens after 15 cycles of salt crystallisation is outlined.

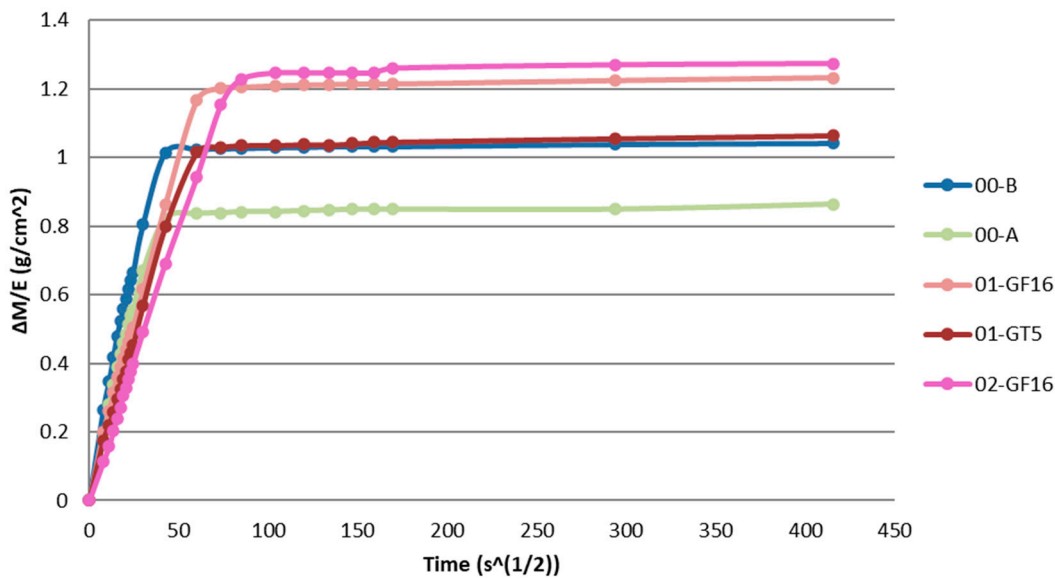

**Figure 11.** Capillarity water absorption curve.

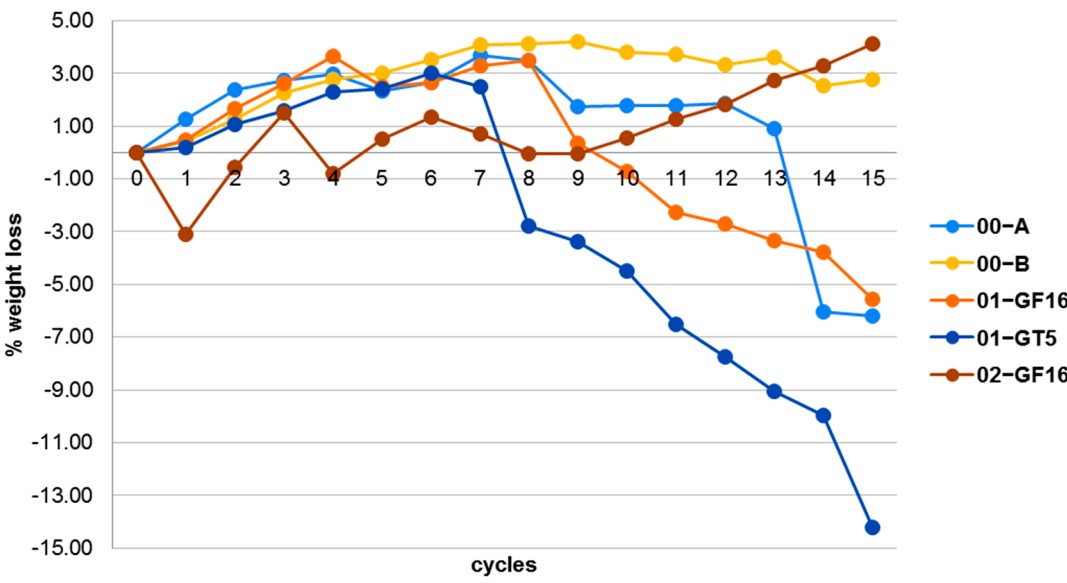

**Figure 12.** Diagram of % weight loss during the 15 cycles of salts crystallization.

**Table 10.** Average value of percentage weight loss among the sound specimen and the specimen in the end of 15 cycles of salt crystallization.

| Name | ΔB % |
|---|---|
| 00-A | −29.65 |
| 00-B | −34.43 |
| 01-GF16 | −28.90 |
| 01-GT5 | −41.93 |
| 02-GF16 | −22.22 |

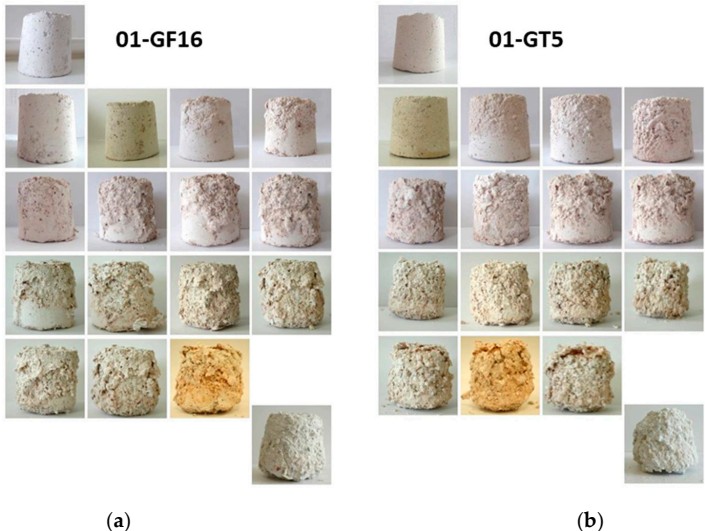

(**a**)                                    (**b**)

**Figure 13.** Photographical documentation of the deterioration of specimens (**a**) 01-GF16 and (**b**) 01-GT5 during the 15 cycles of salt crystallization.

## 4.5. Colour Study

The colour study was focused primarily on the existing colour pallet of the island formed from the geomorphology, the natural landscape with or without building structures, the facades, the roofs and other details of the local identity of Gavdos (Figure 14). Subsequently, the colouration provided from the clay samples was investigated. The clay soils were in the beginning mixed only with water to reveal an intense colour; afterwards, they were used for the production of earthen mortars with earth: marble powder: sand in a 1:1:1 ratio, as well as the production of lime mortars with lime putty: marble powder: sand in a ratio of 1:1.5:1.5 including 10% (*v/v*) earth. In the traditional structures of Gavdos, earthen mortars cannot be used as exterior plaster because they are not water resistant and the typology of the traditional building stock of the island does not offer protection of the wall with a roof overhang. On the other hand, the 10% (*v/v*) of earth in the lime-based mortars offers a very light, almost imperceptible colouration that does not justify its use as a colouring agent in the mortar (as shown in Figure 14).

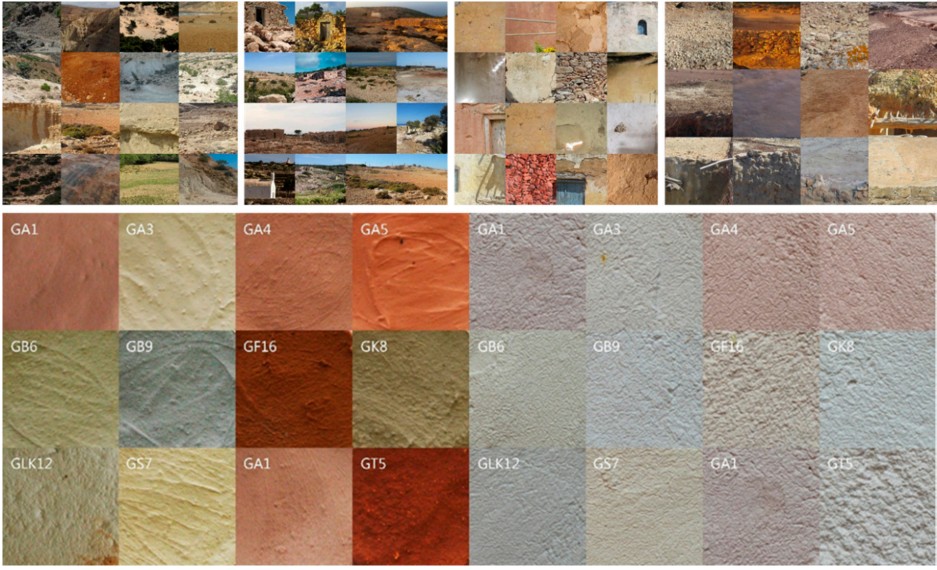

**Figure 14.** Study on the existing colour palette of the island and the colour palette developed from earth-based and lime-based mortars.

Figure 15 illustrates the synthesized limewater (a) and limemilk (b) colours applied to a lime-based substrate. It becomes evident that limewater gives more intense colours and is more transparent, revealing the texture of the underlying surface, whereas limemilk is opaquer. Seven clay soils were chosen to form the colour palette of the present study, namely GA4, GT5, GB6, GS7, GK8, GB9 and GF16. For each clay soil, a mix design for limemilk (A) and a mix design for limewater (B) were developed. The study of limecolours in CIEL*a*b colour scale is presented in Table 11.

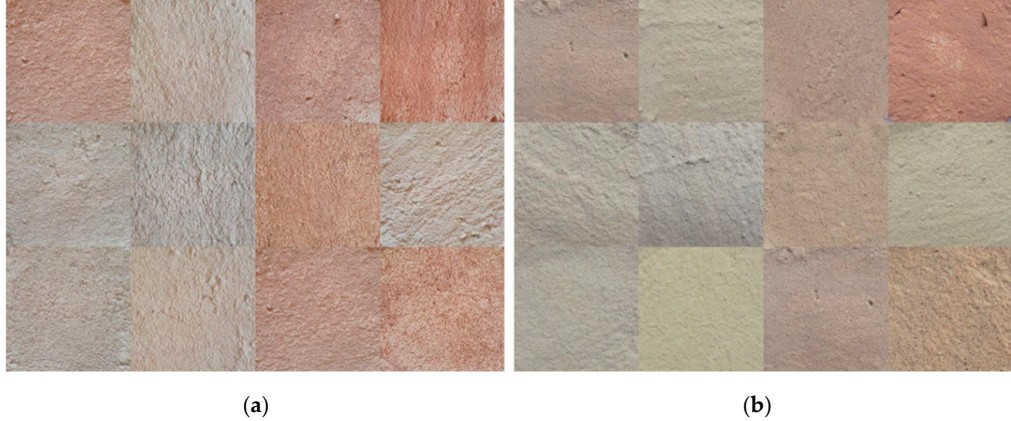

(**a**)          (**b**)

**Figure 15.** Lime colours on lime plaster: (**a**) limewater (1:6); (**b**) limemilk (1:3).

**Table 11.** Limecolours in CIEL*a*b colour scale.

| Name of Sample | L* | a* | b* |
|---|---|---|---|
| GA4-A | 73.18 | 6.50 | 9.12 |
| GB6-A | 78.53 | 0.47 | 8.79 |
| GB9-A | 75.68 | 0.00 | 4.34 |
| GF16-A | 77.89 | 5.43 | 12.96 |
| GK8-A | 79.53 | 1.43 | 9.29 |
| GS7-A | 85.05 | 1.11 | 8.27 |
| GT5-A | 77.94 | 4.48 | 9.91 |
| GA4-B | 69.76 | 7.96 | 10.72 |
| GB6-B | 77.41 | 0.40 | 9.70 |
| GB9-B | 72.95 | −0.20 | 5.09 |
| GF16-B | 73.20 | 6.85 | 16.40 |
| GK8-B | 84.18 | 1.56 | 12.05 |
| GS7-B | 79.57 | 1.79 | 12.19 |
| GT5-B | 78.04 | 6.64 | 14.28 |

### 4.6. Quality Control of Lime-Based Colours with Clay Soils

All the samples that were not placed in the UV chamber, displayed, after a period of 28 days, a colour difference ΔE < 4 (Lightfastness I), according to ASTM D5383 [33]. The samples of limemilk (A) after their placement in the UV chamber for 28 days, displayed, in their majority, a colour difference ΔE < 2 (Lightfastness I), with the exception of sample with pigment from GK8 that displayed colour change 2 < ΔE < 4 and sample with pigment GF16 with colour difference 4 < ΔE < 5 (Lightfastness II). The samples of limewater (B) that were placed in the UV chamber for 28 days displayed a colour difference ΔE < 3 (Lightfastness I), except from sample with GS7 that displayed colour difference in the range of 4 < ΔE < 5 (Lightfastness II). Sample GF16 that presented colour difference above 4 in the case of limemilk, presented colour difference 1 < ΔE < 2 in the case of limewater. Sample GS7 that presented colour difference higher than 4 in the case of limewater, presented colour difference 1 < ΔE < 2 in the case of limemilk. In Figure 16, the colour difference for the specimens with limemilk and limewater in and out of the UV chamber for a period of 28 days is displayed. The measurements were taken once per week. In all the diagrams, we observe that the colour difference of several samples presents an

irregular progress with an increase and decrease in the measurement. These two observations can be attributed to the heterogeneity of the raw materials used.

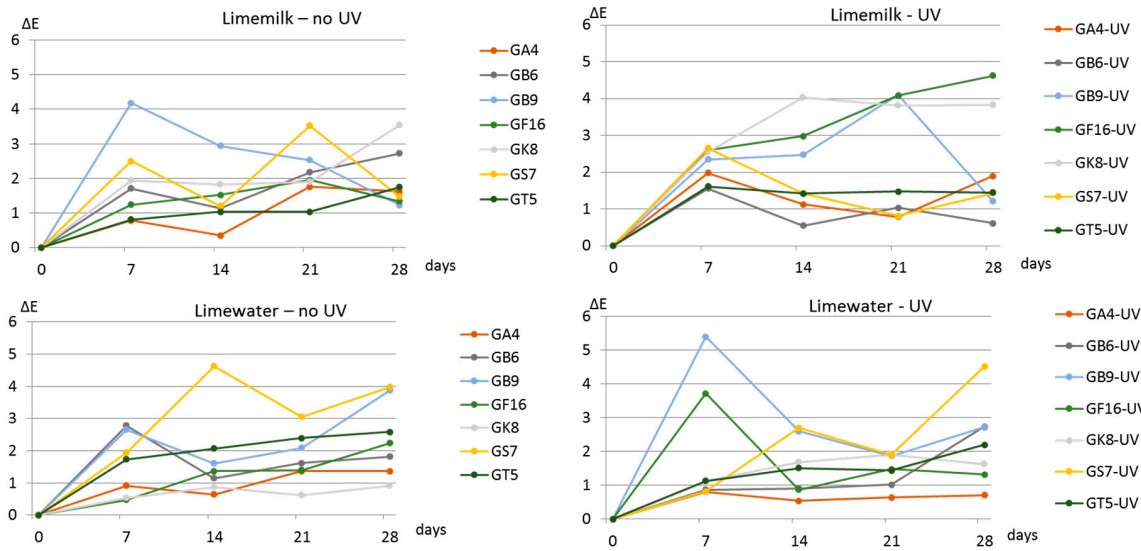

**Figure 16.** Colour difference diagrams for the period of 28 days.

The pigments GB9 and GF16 were used to test the resistance in UV radiation, of limemilk and limewater, respectively, with the addition of natural stabilizers such as, casein, potassium alum, nopal juice and linseed oil. GB9 and GF16 were chosen for their unique colour and chemical composition (Table 12).

**Table 12.** The colour synthesis with natural stabilizers.

| Name | Synthesis |
|---|---|
| GB9-A-01 | Limemilk-GB9-casein |
| GB9-A-02 | Limemilk-GB9-potassium alum |
| GB9-A-03 | Limemilk-GB9-nopal juice |
| GB9-A-04 | Limemilk-GB9-linseed oil |
| GF16-B-01 | Limewater-GF16-casein |
| GF16-B-02 | Limewater-GF16-potassium alum |
| GF16-B-03 | Limewater-GF16-nopal juice |
| GF16-B-04 | Limewater-GF16-linseed oil |

The colour difference results are presented in the diagram of Figure 17 and display performance from very good to excellent. Syntheses with colour difference ΔE > 4 are those with casein and the limewater with potassium alum.

The adhesion (scotch tape) test was performed in specimens with limecolours with and without natural stabilizers. From the implementation of the test, it appears that natural stabilizers improve the adhesion and cohesion ability of colours on the substrate. As is evident in the diagrams in Figure 18, with the addition of natural stabilizers in limecolours, the loss of material falls in all cases below of 0.01 g. The stabilizer that presents the best adhesion performance is linseed oil, with a mean material loss of 0.003 g both for limemilk and limewater.

The results of the abrasion test with the addition of four natural stabilizers are summarized in the diagrams of Figure 19. The higher abrasion resistance is displayed in the limecolours with casein.

Based on the clay soil of the limecolour, for both limemilk and limewater syntheses the resistance to abrasion is presented in the diagram of Figure 20. The worst results in this test are presented for clay soils GS7, GK8 and GT5 that are also the samples with the lower percentage of granules <63 μm with the exception of GK8.

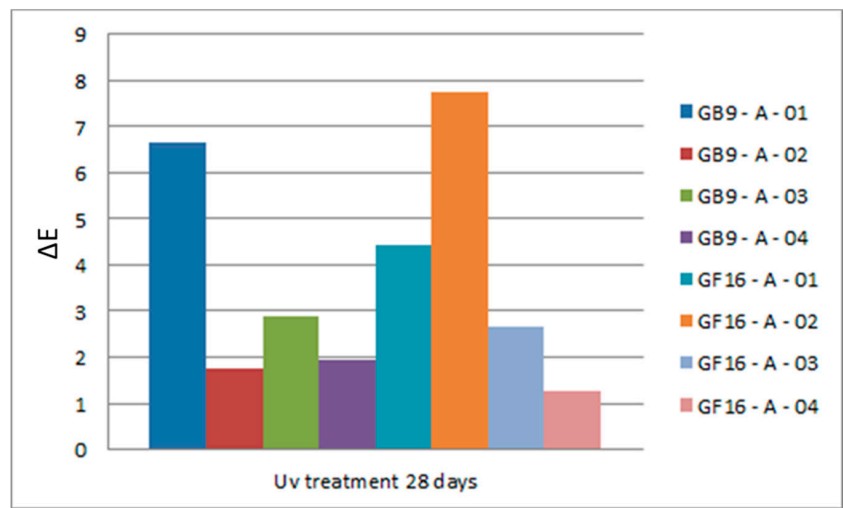

**Figure 17.** Average colour difference of limecolours with natural stabilizers after 28 days in UV radiation.

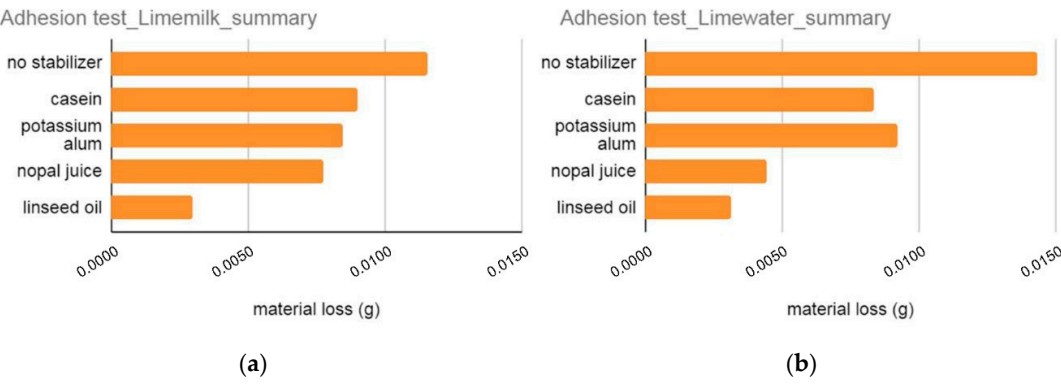

(**a**)  (**b**)

**Figure 18.** Summary of results for adhesion test for (**a**) Limemilk and (**b**) Limewater without stabilizer and with four natural stabilizers.

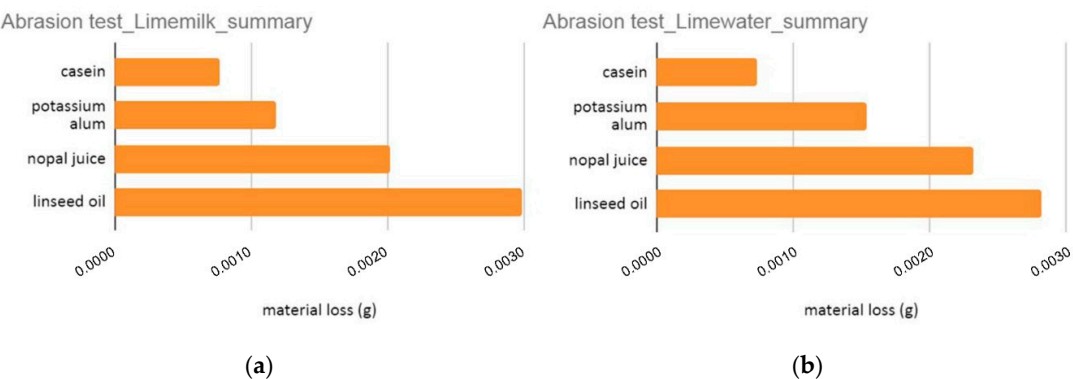

(**a**)  (**b**)

**Figure 19.** Summary of results for abrasion test for (**a**) Limemilk and (**b**) Limewater with four natural stabilizers.

The results of the absorption (sponge) test are summarized in the diagrams of Figure 21. The conclusion derived from this test is that only linseed oil restricts the absorption capacity of limecolours.

The results of the water vapour transmission (cup method) test are presented in the diagram of Figure 22. The samples tested were limemilk (A) and limewater (B) with GA4 and limewater with GA4 casein, potassium alum and linseed oil. All the samples presented a similar behavior of water vapour transmission, which was significantly different from the behavior of the reference sample of silicon.

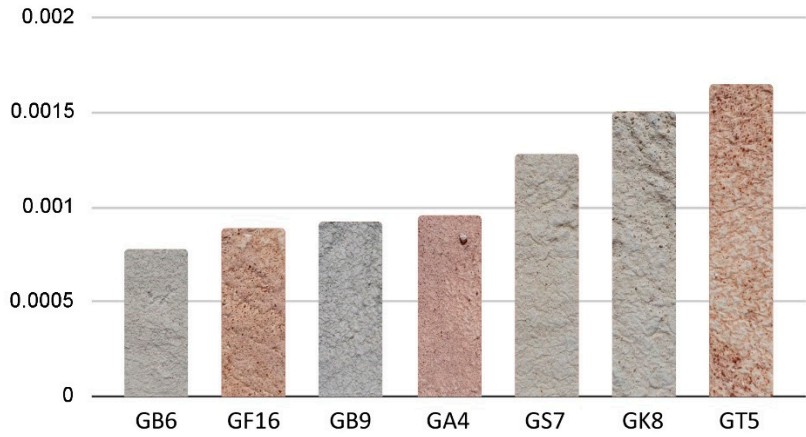

**Figure 20.** Summary of results for abrasion test based on the clay soil (granulometry < 2 mm).

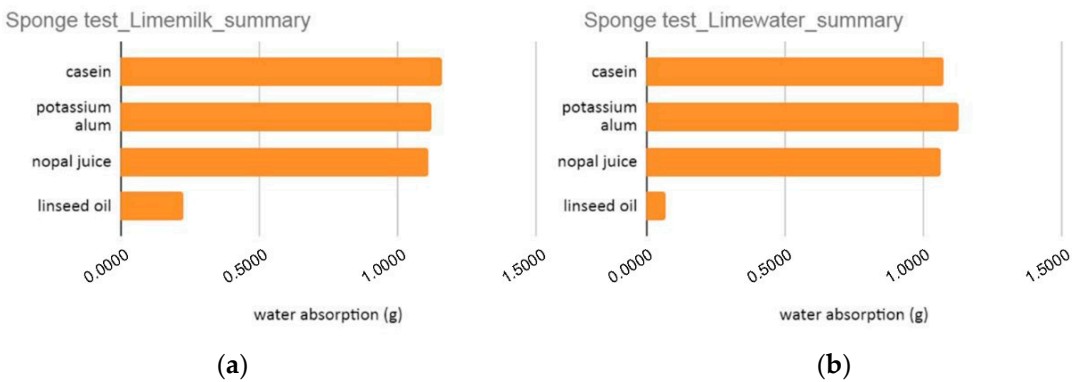

(**a**) (**b**)

**Figure 21.** Summary of results for absorption test for (**a**) Limemilk and (**b**) Limewater with four natural stabilizers.

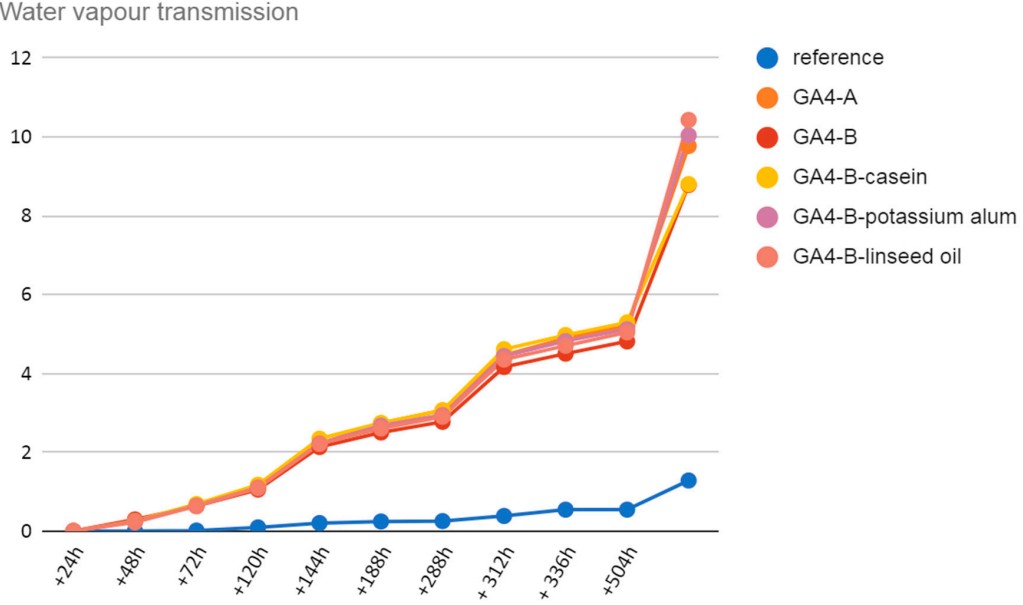

**Figure 22.** Results of water vapour transmission test.

## 5. Discussion

Natural clays can be used as pozzolanic materials when thermally activated. The Luxan test revealed that the pozzolanic activity of each clay sample is related to the calcination temperature, which was differed in each sample. In this study, the thermal treatment of clay soils at a temperature of 600 °C and 700 °C revealed two clay soils that can be used as pozzolanic additives in mortars. In addition to the firing temperature the pozzolanic activity of calcined clays is also enhanced in clays containing low calcite and high kaolinite amounts [2]. The quality control performed on the mortars with calcined clay additives demonstrated positive characteristics for their use in the island of Gavdos with a low shrinkage percentage and durability in marine environment. Further tests are also required for the improvement of the mechanical properties of mortars containing calcined clay. The use of natural and local plastisizers, such as nopal juice, the decrease in the ratio of sand at 1:2, the percentage of calcined clay and the use of finer granulometry, are some factors that can be further studied. The mucilaginous juice extracted from nopal cladodes has been used in ancient Mesoamerica and USA Southwest as an additive in lime mortars and plasters [42]. Although the mechanical properties of the produced mortars are lower than those of the market products, mortars with calcined clay display higher plasticity that make them more compatible with the local limestone.

In addition to their mechanical and physicochemical properties, the colour of these materials, as mineral pigments, was exploited in order to provide a colour palette in harmony with the Mediterranean landscape, inspired by the colour palette of Polygnotus, the ancient Greek painter [43]. Inorganic pigments usually display a better resistance to light and atmospheric conditions [44]. Limecolours with local mineral pigments from clay soil and natural stabilizers displayed a very good performance in the quality control tests and proved resistant to UV radiation, adhesion and abrasion. Simultaneously, they displayed vapour permeable behaviors, suitable for insular environment. Therefore, limecolours provide a very competitive, non-toxic, affordable and ecological alternative for the restoration of traditional and ecological buildings.

Casein, a naturally produced polymer, exhibits excellent adhesive properties [45], potassium alum was by far the most common ingredient for red lake pigments used in western European easel painting from the twelfth until the end of the eighteenth century [46], and linseed oil is one of the most common water repellent additives frequently mentioned in the European literature [47]. Further studies can be implemented for the combined use of natural stabilizers for improvement on the performance of limecolours.

The exploitation of the methodology developed in the framework of this study in other areas with a variety of good quality clay soil is of significant interest. This will lead to the improvement and enrichment of the methodology, the sustainable conservation of traditional building techniques and will also contribute to the creation of a clay soil archive around the world.

## 6. Conclusions

In the framework of this research project, a total of 13 clay soil samples from the island of Gavdos were collected, analysed, thermally treated and tested both as additives in lime-based mortars and as mineral pigments in limecolours. Based on the available results, the following conclusions can be drawn:

The studied clay samples consisted mainly of Quartz, Calcite, Illite, Kaolinite, Chlorite and Albite. Five out of thirteen samples have a clay content of 50% or higher. Only two samples present pozzolanic activity in firing temperatures of 600 °C and 700 °C. Both of them comprise a low calcite content (9% or lower) and a high clay content (70% or higher), the one with the highest clay content (83.7%) also exhibits the highest percentage of Kaolinite (50%). The negative conductivity measurements in samples during the Luxan test are attributed to their considerable calcite content (more than 20%).

The lime mortars with calcined clay from Gavdos display a moderate mechanical performance to be used as joint mortars. However, their superior performance in plasticity, capillarity water absorption

and resistance to salt crystallization than the commercial products, establish the designed mortars as candidates for effective plasters and renders.

The unfired clay soil used as mineral pigment in limecolours demonstrated a very good to excellent performance in UV radiation. With the addition of natural stabilizers, the limecolours also displayed very satisfying results for UV radiation, adhesion, abrasion and absorption, without affecting the water vapour transmission capacity of the samples.

There is a great potential in the exploitation of local raw material from the island of Gavdos for the restoration of the traditional building stock on the island in terms of resource efficiency, environmental impact and preservation of the local identity. In addition, the methodology presented in this study can be established as a powerful tool of discriminating effective clayey soils that, upon low heating treatment, can be used as pozzolanic additions in mortars and plasters.

**Author Contributions:** Conceptualization, P.-N.M. and C.O.; methodology, P.-N.M., C.O. and A.F.; validation, P.-N.M., V.P. and N.K.-K.; formal analysis, A.F.; investigation, C.O. and A.F.; resources, P.-N.M.; data curation, A.F. and K.K.; writing—original draft preparation, A.F.; writing—review and editing, P.-N.M. and A.F.; visualization, A.F. and C.O.; supervision, P.-N.M.; project administration, P.-N.M.; funding acquisition, P.-N.M. All authors have read and agreed to the published version of the manuscript.

**Funding:** This research was funded by the Hellenic General Secretariat for Investment and Development, grant number 26695/06-02-2017.

**Acknowledgments:** The authors greatly acknowledge Apostolos Alexopoulos from the Faculty of Geology and Geoenvironment of the National & Kapodistrian University of Athens and the Member of the municipal council of Gavdos, ex-mayor, ceramist, Gelly Kallinikou for their contribution in the identification of clay soils of Gavdos. Architects Anna-Maria Lagou and Panagiota Dania are also acknowledged for their contribution in part of the experiments of this project.

**Conflicts of Interest:** The authors declare no conflict of interest. The funders had no role in the design of the study; in the collection, analyses, or interpretation of data; in the writing of the manuscript, or in the decision to publish the results.

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
