# Peer review of "Ecological Restoration Plasters and Mineral Pigments Designed with Raw Material from the Island of Gavdos†"

_infrastructures, doi:10.3390/infrastructures5120110_

Round 1

Reviewer 1 Report

The paper is quite interesting and well written.

there is something to revise in the presentation of the set of specimen tested, and it should be clarified.  Currently it is not clear how many specimen were involved in the overall research and how they were composed. Think about adding a figure presenting the overall specimen set.

more in depth Discussion is needed.

394 = please check the use of the term “hydrophobicity” which seems to me extraneous to the topic. You’re measuring water absorption not hydrophobicity, which could be estimated through the contact angle test. Glass is not absorbing at all, but hydrophilic.

Fig. 14 is not clear. Please discuss the variability of total color difference ΔE during the cycles

Some minor revisions are as follows

Line 34 = 21 nm

Line 52 = substitute depended with dependent

117 = non breathable / non permeable ? or water vapor permeable?

161 = please explain better what do you mean with water repellency in this context

Table 1: do you think a legend about GF16 700 and GT5 600 is needed? Ref to Fig. 5 ?

191 = following… followed ; resolve the repetition

199 = Over oven

Please clarify the preparation of limecolors. What is the total number of different limecolors prepared? It seems to me that in Table 3 the “binder” formulation is described (10 different binder from A0 to B4), then each type of binder was tested with the single specific 7 clay soils (line 246) leading to a total of 70 different limecolors ? Am I right? Please clarify.

Could you please provide some more detail about the quantitative method adopted in XRD analyses?

Fig. 10 Check Pouzzolanic

385 = Young not young

Fig 15 please substitute δ with Δ

549 antche? Is that right?

Reviewer 2 Report

Very interesting read. Sound scientific and technical study with good reporting of the results.

Just minor checks mainly in the introduction:

Line 34 – check units (21nm)

Line 62 – you cannot state “recently increased attention” based on an article from 2001 that is in itself a review of past paper (older that 2001).

Line 73 – it seems odd that you mention mural paintings in Iraq and Syria and don’t mention Knossos. Maybe a reference should be included for those murals.

Line 74 – Syria, not Suria

Line 93 – 14 m in height

Line 95 – countryside

Line 171 – either “in order to prevent future interventions” or in order not to compromise future interventions” Which one do you mean?

I propose that the chapter “Results” changes its name to “Results and discussion” (one of the options of this journal) and that the text under discussion may be merged with the Conclusions.

Reviewer 3 Report

This paper is a useful experimental study that shows the possibilities of the use of mineral pigments from Gavdos as puzzolanic material. However, before acceptance requires mayor changes, whose suggestions are described below:

  1. Energy efficiency of clay renders is not justified.
  2. Page 2, line 92. Reference to Light House is unnecessary.
  3. Page 5, line 199. Oven instead of over.
  4. Page 6, line 224. Difference between limewater and limemilk is not explained.
  5. Figure 1 corresponds to Results, not to Material and Methods.
  6. This research tries to make a clay file archive used as pigments. Color is important, then a color scale, CIELAB or CIELCH, could be used to define numerally the color of clays and mortars.
  7. Figure 3. Caption is incorrect, maybe UV and Dark Chamber?
  8. Page 8, line 272. What is the precision of the scale?
  9. Page 9, line 285. Reference [37] corresponds to adhesion by tape test, not to absorption test.
  10. Figure 8. Caption does not indicate the name of the specimen included in figure.
  11. Figure 9. The material that undergoes a transformation in each peak could be indicated in the graph.
  12. Figure 10. Selection showed in figure is not explained in text.
  13. Table 8. Young's module values are dubious. There is a lot of dispersion of results, in some cases the samples acquire resistance with time but lose young modulus. This fact must be clarified and explained.
  14. Table 10. The abbreviation ΔB used in the table is not used in the text. Weight loss?
  15. Figure 15. The y-axis value is incorrect, is ΔE.
  16. The discussion is very poor and there are no references to the results of other researches. It should be expanded to become a paper with scientific interest.
  17. Nomenclature of specimens should not be used in conclusions, it is better to refer to their composition.

Reviewer 4 Report

The article provides an interesting insight into the issue of restoration plasters, on the example of traditional Gavdos architecture. A wide variety of tests was performed, all connected to the practical use of the lime-pozzolan mortars developed by the Authors. The investigation is thorough and systematic. However, there are some issues:

  1. Table 1 - "GF" should be explained. It is the first time it appears in the text, and it is not clear what it is. While it is explained later, it is confusing,
  2. Are Authors absolutely sure that the sample size for strength testing according to EN-1015 is 50x50x50mm and not 40x40x80mm?
  3. Moreover, are Authors sure that the curing of the samples with aerial lime was conducted according to standard EN-1015-11? If not, could the authors give a comment on that?
  4. In majority of the data presented, there is a lack of statistical analysis. It is also hard to ascertain how many samples were used in what test.

Round 2

Reviewer 1 Report

I suggest just the following change:

Line 121: replace "water vapor impermeable" with "low water vapor permeable"

Reviewer 3 Report

All recommendations have been attended